

# Global sensitivity analysis of GEOS-Chem modeled ozone and hydrogen oxides during the INTEX campaigns

Kenneth E. Christian[1], William H. Brune[1], Jingqiu Mao[2], and Xinrong Ren[3,4]

[1]Department of Meteorology and Atmospheric Science, Pennsylvania State University, University Park, PA (USA)
[2]Department of Chemistry and Biochemistry and Geophysical Institute, University of Alaska at Fairbanks, Fairbanks, AK (USA)
[3]Department of Atmospheric and Oceanic Science, University of Maryland, College Park, MD (USA)
[4]Air Resources Laboratory, National Oceanic and Atmospheric Administration, College Park, MD (USA)

*Correspondence to:* Kenneth Christian (kec5366@psu.edu)

**Abstract.** Making sense of modeled atmospheric composition requires not just comparison to in situ measurements, but also knowing and quantifying the sensitivity of the model to its input factors. Using a global sensitivity method involving the simultaneous perturbation of many chemical transport model input factors, we find the model uncertainty for ozone ($O_3$), hydroxyl radical (OH), and hydroperoxyl radical ($HO_2$) mixing ratios and apportion this uncertainty to specific model inputs for the DC-8 flight tracks corresponding to the NASA INTEX campaigns of 2004 and 2006. In general, when uncertainties in modeled and measured quantities are accounted for, we find agreement between modeled and measured oxidant mixing ratios with the exception of ozone during the Houston flights of the INTEX-B campaign and $HO_2$ for the flights over the northernmost Pacific Ocean during INTEX-B. For ozone and OH, modeled mixing ratios were most sensitive to a bevy of emissions, notably lightning $NO_x$, various surface $NO_x$ sources, and isoprene. $HO_2$ mixing ratios were most sensitive to CO and isoprene emissions as well as the aerosol uptake of $HO_2$. With ozone and OH being generally over predicted by the model, we find better agreement between modeled and measured vertical profiles when reducing $NO_x$ emissions from surface as well as lightning sources.

## 1 Introduction

Air quality and atmospheric composition for the United States and North American continent is at an intersection between competing drivers. On one hand, emissions controls and cleaner burning fuel sources have resulted in a significant decrease in US $NO_x$ ($NO_x \equiv NO + NO_2$) emissions (e.g., de Gouw et al., 2014). On the other, for many locations, especially in the western US, air quality has not improved proportionally to these emissions reductions, in part due to transport from Asia (Verstraeten et al., 2015; Lin et al., 2017). Thus better understanding the complicated processes that govern atmospheric composition for North America is vital in making informed regulatory decisions.

Correctly modeling atmospheric composition is a difficult endeavor, but one of great importance. Oxidants are of particular interest and importance when it comes to tropospheric chemical modeling and applications relating to both health and climate change including ozone, which has deleterious environmental and human health effects, and the hydroxyl radical (OH),



which largely determines the lifetimes of volatile organic compounds (VOCs) and greenhouse gases like carbon monoxide and methane. Thus, in trying to understand current and future air chemical processes, oxidants are a worthy place to start.

Modeling the composition of the atmosphere is complicated, notwithstanding the fact that model inputs, such as emissions, chemical reactions, and transport are not perfectly understood and cannot be perfectly represented in computer models. To make sense of these shortcomings, sensitivity and uncertainty analyses are useful tools in both determining the robustness of modeled results and identifying and quantifying sources of error. Generally, sensitivity analyses fall into two main camps: local and global. Local sensitivity analyses involve the perturbation of individual model inputs one at a time over a prescribed segment of the input space. Global sensitivity analyses, however, feature the simultaneous perturbation of multiple inputs across the breadth of their uncertainty ranges (Rabitz and Aliş, 1999; Saltelli et al., 2008). The advantage of these simultaneous perturbations is that non-linear interactions between model factors are taken into account in global sensitivity analysis, an important advantage considering the non-linear nature of the interactions between emissions, chemistry, and meteorology that underlie atmospheric composition modeling.

With the computationally expensive nature of running chemical transport models (CTMs) such as the GEOS-Chem (Goddard Earth Observing System-Chemistry) model used in this study, global sensitivity methods, which require hundreds of model runs to provide meaningful statistical results, have been unsurprisingly lacking from the literature save for a some recent work (Brewer et al., 2017; Christian et al., 2017). Instead, the sensitivity analyses of GEOS-Chem modeled results has either used local methods in which the factor of interest is perturbed individually and compared to the model state without this perturbation, or the GEOS-Chem adjoint (Henze et al., 2007). This type of analysis has been completed for a variety of emissions (Fiore et al., 2002; Guerova et al., 2006; Jaeglé et al., 2005; Mao et al., 2013b; Fiore et al., 2005; Qu et al., 2017), meteorological (Wu et al., 2007; Heald et al., 2010), and chemical factors (Mao et al., 2013a; Newsome and Evans, 2017). While useful in determining some individual sensitivities, these methods neither can nor were intentioned to provide a complete picture of model sensitivities in which many inputs have uncertainties.

To gain a better grasp of air chemical processes over North America, and the regions both up and downwind of the continent, various academic and governmental entities took part in the NASA sponsored Intercontinental Transport Experiment (INTEX) campaigns, part of the International Consortium on Atmospheric Transport and Transformation (ICARTT). The INTEX-NA (INTEX-North America) part of the ICARTT campaign took place in two phases: INTEX-A (summer 2004) and INTEX-B (spring 2006). The INTEX-A campaign sought to characterize the air chemistry of eastern and central United States and Canada and was based out of Pease Air National Guard Base in Portsmouth, New Hampshire and Mid America Airport/Scott Air Force Base in Western Illinois (St. Louis, Missouri metropolitan area). After INTEX-A, which characterized the air composition of the continent, INTEX-B sought to study both the North Pacific background and Asian outflow, and Mexican outflow over the Gulf of Mexico. These flights were based out of Houston, Texas; Honolulu, Hawaii; and Anchorage, Alaska.

Through a global sensitivity analysis of modeled oxidants during INTEX we aim to meet a few goals. One, determine the uncertainty in modeled results arising from uncertainty in the model inputs. Two, determine which of these inputs are most responsible for the uncertainty in the modeled results. Three, determine which perturbations to the model allow for a better match to in-situ observations collected during the campaigns. In allowing for the calculation of model uncertainties





and sensitivities to many input factors, a global sensitivity analysis is well suited for these objectives. Knowing the model sensitivities will provide direction not only for future model improvements but also for identifying the most impactful directions for future research.

## 2 Methods

In the following section, we briefly describe the methods employed in this study. For a more detailed description, please refer to Christian et al. (2017).

### 2.1 Model

We use in this study the standard GEOS-Chem model (v9-02), a popular global chemical transport model (Bey et al., 2001). There are a few different resolutions available to modelers, but to facilitate the construction of our sensitivity ensemble, we

used the coarser horizontal resolution of $4° \times 5°$. Model resolution is an important consideration for chemical transport models, but the errors associated with resolution choices are usually less than those coming from chemistry, meteorology, and emissions (Wild and Prather, 2006). In general, there were typically small differences between modeled results using either $4° \times 5°$ or $2° \times 2.5°$ resolutions but we illustrate in our results where this is not the case.

Our GEOS-Chem model runs were driven by the Modern-Era Retrospective Analysis for Research (MERRA) meteorological

model for INTEX-A, while the INTEX-B model runs were driven by GEOS-5 (Goddard Earth Observing System). This difference is due to GEOS-5 model availability not extending far enough back in time to facilitate its inclusion in the INTEX-A runs. When comparing modeled results for INTEX-B running MERRA, there were extremely small differences between the modeled results using either meteorological model. As uncertainties are not published for the meteorological models, we define our meteorological uncertainties as the average of the monthly standard deviations of the difference between GEOS-4

and GEOS-5 meteorological fields for 2005, a year of overlap between the models.

Generally, the model ensemble made use of the standard emissions inventories. For much of the developed world, including much of North America, Europe, and East Asia, the regional emissions inventories overwrote the default Emission Database for Global Atmospheric Research (EDGAR) or REanalysis of the TROpospheric chemical composition (RETRO) fields. Lightning $NO_x$ is treated through the scheme of Price and Rind (1992) with close to a factor of 2 greater lightning $NO_x$ yield over

the midlatitudes compared to the tropics (500 mol flash$^{-1}$ vs 260 mol flash$^{-1}$). The differential between the treatment of tropical and midlatitudinal $NO_x$ yields was created to match observations (Huntrieser et al., 2007, 2008; Hudman et al., 2007). Recent research has questioned the arbitrary boundary in lightning $NO_x$ yields and show the sensitivity of regions around the tropical/midlatitude boundary to this treatment (Zhang et al., 2014; Travis et al., 2016). We show in our results where this is a consideration. Transport of stratospheric ozone into the troposphere is parameterized by the Synoz algorithm (McLinden et al.,

2000) in which 500 TGyr$^{-1}$ of ozone is advected through the tropopause.

Uncertainties in emissions in this study ranged from factors of 2 to 3 with higher uncertainties in biomass and soils emissions. This higher uncertainty is due to the wide range of values in the literature, (e.g., Jaeglé et al., 2005; Schumann and Huntrieser,





2007; Vinken et al., 2014). We assume uncertainties of a factor of 2 for lightning $NO_x$ (Liaskos et al., 2015), biogenic VOC (Guenther et al., 2012), stratospheric-tropospheric exchange of ozone, default and regional anthropogenic, ship, and methyl bromoform emissions.

Chemical rate uncertainties were found from NASA's Jet Propulsion Laboratory's (JPL's) evaluation (Sander et al., 2011).

For the most part, chemical rate uncertainties are lower than those of emissions inventories, around 20-30% for many chemical kinetic and photolysis rates. Uncertainty in the rate of aerosol particle uptake of the hydroperoxyl radical ($HO_2$) (gamma $HO_2$) was assumed to be a factor of 3. In the case of gamma $HO_2$, we use the standard model treatment in which $\gamma_{HO_2} = 0.2$ (Jacob, 2000) and yields $H_2O$, a terminal $HO_x$ ($HO_x \equiv OH + HO_2$) reaction (Mao et al., 2013a). Not only is there uncertainty in the rate of this uptake, but there is also uncertainty in the product of this reaction, and whether or not $H_2O_2$ is produced instead of

or alongside $H_2O$. In this study, we generally find small differences between these possibilities.

## 2.2   Global Sensitivity Analysis

The Random Sampling-High Dimensional Model Representation (RS-HDMR) (Rabitz and Aliş, 1999; Li et al., 2001) is a global sensitivity analysis method used in conjunction with other air chemistry studies (Chen and Brune, 2012; Chen et al., 2012; Christian et al., 2017). The method involves the simultaneous perturbation of model factors across their respective

uncertainties. Instead of randomly sampling the input space as prescribed, we sample using a quasi-random number sequence (Sobol, 1976). Quasi-random sampling allows for a more efficient sampling of the input space facilitating reliable results with fewer runs. Following common practice, we discarded a set of initial values when creating the quasi-random sequence, in our case the first 512, as a spin up.

Previous sensitivity analyses implementing the HDMR method or its variations often use thousands of model runs. With

CTMs like GEOS-Chem, this computational cost is prohibitive. Instead, we limit our ensemble to 512 model runs. As seen in Lu et al. (2013) and this study, we find the sensitivity results to converge after a few hundred runs supplying confidence in the indices calculated here.

Conceptually, the HDMR method describes the modeled output as a collection of polynomials relating the model output to the inputs, both individually and collectively.

$$f(x) = f_0 + \sum_{i=1}^{n} f_i(x_j) + \sum_{1 \le i \le n} f_{ij}(x_i, x_j) + ... + f_{12...n}(x_1, ..., x_n) \tag{1}$$

Here $f_0$ is the zeroth order component, a constant equivalent to the mean (Eq. (2)), $f_i$ is the first order effect corresponding to the independent effect of the input $x_i$ on the output (Eq. (3)), $f_{ij}$ corresponding to the second order effect on the output of inputs $x_i$ and $x_j$ working cooperatively (Eq. (4)), on down to the $n^{th}$ order effect on the output by all the inputs working cooperatively (Rabitz and Aliş, 1999).

$$f_0 \approx \frac{1}{N} \sum_{s=1}^{N} f(x^s) \tag{2}$$



$$f_i(x_i) \approx \sum_{r=1}^{k_i} \alpha_r^i \varphi_r^i(x_i) \tag{3}$$

$$f_{ij}(x_i, x_j) \approx \sum_{p=1}^{l_i} \sum_{q=1}^{l_j} \beta_{pq}^{ij} \varphi_p^i(x_i) \varphi_q^j(x_j) \tag{4}$$

Here $\varphi$ represents orthonormal polynomials, $k_i$, $l_i$, and $l_j$ represent the orders of the polynomials, $\alpha$ and $\beta$ are constant coefficients.

With each component function being orthogonal, the total variance can be split into a summation of the variances of all the polynomials in Eqs. (3) and (4) (Li et al., 2010). For example:

$$V(f(x)) = \sum_{i=1}^{n} V(f_i(x_i)) + \sum_{1 \leq i \leq n} V(f_{ij}(x_i, x_j)) + ... + V(f_{12...n}(x_1, ..., x_n)) \tag{5}$$

Where $V(f_i(x_i))$ represents the variance of the first order effect due to the input $x_i$ and so forth. Normalizing the individual variances in Eq. (5) by the total variance results in the creation of sensitivity indices for each input (Eq. (6)). While sensitivity indices can similarly be found for the functions relating to the second and higher order interactions between inputs, these indices need more model runs than presented here for meaningful results.

$$S_i = \frac{V(f_i(x_i))}{V(f(x)))} \tag{6}$$

To focus the RS-HDMR analysis on the most important model inputs, we completed a preliminary Morris method sensitivity test (Morris, 1991) for both the INTEX-A and INTEX-B domains, including any factor within around 15 % of the most sensitive factor for ozone, OH, or $HO_2$. Using the Morris Method as a preliminary step in RS-HDMR tests is a common practice in multiple RS-HDMR sensitivity studies (Ziehn et al., 2009; Chen et al., 2012; Lu et al., 2013). This resulted in 39 factors being included in the RS-HDMR analysis for INTEX-A and 47 for INTEX-B (Tables 1 and 2 respectively).

**2.2.1   Uncertainties**

Before perturbing the inputs and running the model, the next step was to create the uncertainty distributions for the prescreened model inputs using the uncertainties listed earlier in the methods section and in Tables 1 and 2. For the majority of the factors, we used lognormal uncertainty distributions where the standard deviations were determined by $\sigma = (f-1/f)/2$ (Gao et al., 1995; Yang et al., 1995) where f is the published uncertainty factor. Normal distributions were used for some meteorological factors
(relative and specific humidity, soil wetness, and temperature). To allow model perturbations time to spread globally, all runs in the model ensemble were spun up 9 months before the first flight for the respective campaigns.

**2.2.2   Calculation of sensitivity indices**

RS-HDMR sensitivity indices were calculated using graphical user interface - HDMR (GUI-HDMR), a free MATLAB package (http://www.gui-hdmr.de) (Ziehn and Tomlin, 2009). As in Christian et al. (2017), in running GUI-HDMR, the inputs were





scaled according to their percentiles within their respective uncertainty distributions and the correlation method option was applied (Kalos and Whitlock, 1986; Li et al., 2003).

## 2.3 Measurements

The NASA DC-8 carried a suite of state of the science instruments during both INTEX-A and INTEX-B (Singh et al., 2006, 2009). For comparison to the modeled $HO_x$ mixing ratios, we compare to the measurements taken by Pennsylvania State University's Airborne Tropospheric Hydrogen Oxides Sensor (ATHOS) (Brune et al., 1998). In this instrument, $HO_x$ is measured using laser-induced fluorescence (LIF). Ozone mixing ratios were measured by NASA-LaRC (Langley Research Center) using nitric oxide chemiluminescence (Weinheimer et al., 1994).

Interferences in OH and $HO_2$ measurements are a concern with ATHOS and other measurement techniques (Ren et al., 2004; Fuchs et al., 2011; Mao et al., 2012). Typically these interferences are less than a factor of 2 for $HO_2$ and between a factor of 1.2 and 3 for OH. Interferences in OH and $HO_2$ are mostly a concern in the boundary layer above forested or urban environments as they occur in the presence of alkenes or aromatics. For much of the mid to upper troposphere and the marine domains sampled in much of INTEX-B, these interferences will be negligible.

## 2.4 Box Model

As an additional comparison to both the chemical transport model and the measurements, we also analyze oxidant mixing ratios calculated by a time dependent zero dimensional box model. In this modeling approach, $HO_x$ mixing ratios are calculated using a model constrained by other trace gas measurements measured aboard the DC-8 and is integrated until the box model reaches a consistent diurnal steady state. At a minimum, the model is constrained by ozone, CO, $NO_2$, non-methane hydrocarbons, acetone, methanol, temperature, dew and frost point of water, pressure, and calculated photolysis frequencies (Ren et al., 2008). These model calculations are available alongside the measurements in the NASA Langley archives for the campaigns. For a more detailed description of the box model, please refer to Crawford et al. (1999); Olson et al. (2004); Ren et al. (2008).

## 2.5 Comparison of modeled and measured results

In order to compare the measurements along the DC-8 flight track to GEOS-Chem, the Planeflight option was used allowing for modeled quantities to be output in one-minute intervals along the model flight track. With a relatively coarse horizontal resolution chosen, it is a concern that GEOS-Chem would miss meso to synoptic scale features that could be important for correctly modeling oxidant abundances. With our analysis averaging over many flights, many of these differences would be averaged out.

# 3 Results

During INTEX-A, the NASA DC-8 primarily sampled the eastern half of the United States and Canada INTEX-A during the summer of 2004. In contrast to the mostly continental study area of INTEX-A, INTEX-B largely took place over the Gulf of





Mexico and the North Pacific (Fig. 1) in the spring of 2006. In both campaigns, the aircraft sampled the troposphere at a variety of altitudes from the surface to near the tropopause (bar graphs in Figs. 2, 3, 4, and 5). In INTEX-B, the results are split into three separate domains outlined in Fig. 1 and named according to the city in which the flights were based: Houston, Texas; Honolulu, Hawaii; and Anchorage, Alaska.

## 3.1 Uncertainty

### 3.1.1 INTEX-A

For ozone and OH, GEOS-Chem modeled mixing ratios were consistently higher than measurements (Fig. 2). Throughout the vertical column, GEOS-Chem modeled ozone was around 10 ppb greater than measurements. For OH, modeled and measured values were similar close to the surface, but the disagreement widens higher, with modeled values being a factor of $\sim$1.6 greater than measurements around 6 km. Unlike GEOS-Chem, the box model generally agreed with the measured OH profiles suggesting that the model errors for OH are likely arising outside of the chemical mechanism, such as emissions sources. In contrast to ozone and OH, measured $HO_2$ profiles were generally greater than the model ensemble with the widest disagreement coming close to the surface. Unlike OH, $HO_2$ profiles modeled by the box model generally agreed with GEOS-Chem more than they did the measurements. This model-model agreement suggests that either the model errors may be arising from the largely similar chemistry of the two models or the measurements are incorrect, perhaps due to peroxy radical interference. The agreement between GEOS-Chem and ATHOS $HO_x$ profiles presented here is different than Hudman et al. (2007) due to errors found in the calibration of the measurements (Ren et al., 2008).

Part of this disagreement in mixing ratios could be attributed to uncertainties in the modeled values. We find 1 $\sigma$ uncertainties for the modeled oxidant mixing ratios to range from 19-23 % for ozone, 27-36 % for OH, and 18-37 % for $HO_2$ in the different vertical bins. When taking into account both uncertainties in model input factors and measurements, we find there to be overlap between all the oxidant profiles. This overlap shows that the uncertainties in the model and measurements can explain the difference between the model and measured profiles.

### 3.1.2 INTEX-B Houston

The vertical profiles for ozone, OH, and $HO_2$ all follow a similar pattern: general agreement between measured and modeled mixing ratios near the surface turning to model overestimation above 4 km or so (Fig. 3). In the case of ozone, the model-measurement gap persists even when accounting for measurement uncertainty, especially from 5 km higher. As a consequence of this model overprediction of ozone, OH and $HO_2$ both are also overpredicted by GEOS-Chem above 4-5 km, but unlike ozone there is overlap at all levels between the measured and modeled values when uncertainties in both are taken into account. Generally, there are small differences between the median of the 4° x 5° model ensemble and a finer resolution 2° x 2.5° run, however, there are some larger differences between these two runs, with ozone mixing ratios being reduced by 7-9 ppb above 5 km in the finer resolution. Conversely, below 5 km, the finer resolution run produces higher OH mixing ratios (about 0.06 ppt or $\sim$30 % higher), roughly on the order of the 1 $\sigma$ model uncertainty.





Unlike GEOS-Chem, the box model tended to better agree with measurements higher in the troposphere for OH (Fig. 3). In the case of OH mixing ratios, the box model was around a factor of 2 greater than measurements in the first vertical bin and around 30 % greater up through 4 km. Higher than 4 km, the box model and measurements largely agreed. For HO$_2$ mixing ratios, the box model was greater than observations at all heights but was marginally closer than GEOS-Chem to the measured profile.

Model ozone uncertainty was largely altitude independent, running between 19 and 21 % below 8 km. Uncertainty in modeled OH was between 28 and 40 % with uncertainty on a percentage basis ranging highest near the surface and above 7 km (Fig. 3). Model HO$_2$ uncertainty followed a similar vertical pattern to OH with the highest uncertainty coming near the surface (~30 %) and lower in the middle troposphere (18-20 % from 3 km up through 8 km).

### 3.1.3 INTEX-B Honolulu

Vertically, uncertainty in ozone is nearly altitude independent, ranging between 17.5 and 20.5 % (1 $\sigma$) (Fig. 4). While GEOS-Chem on average comes close to the average measured values, the model fails in matching the measured profile shape. Near the surface, the GEOS-Chem is around 12 ppb less than measured values. This underprediction shifts to overprediction around 4 km with the model overpredicting 25-30 ppb around 9-10 km. This under and overprediction by the model at low and high altitudes is outside the model and measurement uncertainties.

In contrast to ozone, the uncertainty in OH mixing ratios is high and vertically variable (Fig. 4). From 0-3 km, uncertainty is roughly around 32-36 % before increasing through the middle troposphere to 38-40 %. For all altitudes, measured and model values were within each other's uncertainty range. The box model agreed well with OH measured mixing ratios, especially above 5 km with more modest agreement and slight overprediction below.

Compared to OH, uncertainty in HO$_2$ mixing ratios is lower but follows the same pattern of increasing with altitude (Fig. 4). We find uncertainty rising from 16-20 % between the surface and 4 km, to between 23-30 % from 5 km higher. Generally, GEOS-Chem replicated the measured HO$_2$ mixing ratio profile within a couple ppt. Like OH, the box model generally agreed well with measured HO$_2$ mixing ratios. The overall agreement between the oxidant profiles in this domain may be attributable to the reduced surface emissions sources in this remote, Central Pacific domain.

### 3.1.4 INTEX-B Anchorage

In contrast to the previous regions analyzed here, measured ozone, OH, and HO$_2$ mixing ratios were generally greater than GEOS-Chem modeled values in nearly every vertical bin (Fig. 5). Ozone mixing ratios were underpredicted by the model around 10 ppb, with the difference between modeled and measured values maxing out at 17 ppb around 4 km. Except for near the surface where the model was around 0.04 ppt too high and above 8 km, GEOS-Chem generally underrepresented OH by a couple hundredths of a ppt. These differences are within the model and measurement uncertainty. HO$_2$ mixing ratios showed some of the widest disagreement between modeled and measured values with the model being anywhere from a 1.6 ppt short near the surface to upwards of 6.8 ppt between 3 and 4 km.





Compared to GEOS-Chem, the box model performs better in matching the measured OH and $HO_2$ mixing ratio profiles. In particular, while still somewhat underpredicting $HO_2$ mixing ratios, the box model does match the shape of the measured $HO_2$ profile unlike GEOS-Chem (Fig. 5). Because of this relatively close match between the box model and the measurements, the disagreement between GEOS-Chem and the measurements could be arising outside of the chemical kinetics. Conversely, the

box model may be better matching the measured profile just due to its lack of aerosol uptake of $HO_2$. In the Arctic, the aerosol uptake to $HO_2$ is a major loss pathway for $HO_2$ (Whalley et al., 2015). Without this loss pathway, the box model may have artificially high $HO_2$ mixing ratios.

Uncertainty in modeled ozone mixing ratios was relatively low, ranging between 13 and 20 %. In contrast, uncertainty in both OH and $HO_2$ mixing ratios were considerable ranging between 34 and 57 % for OH and 21 and 40 % for $HO_2$ (Fig. 5).

This higher uncertainty is in part a product of the very low mixing ratios modeled in this northern domain with OH mixing ratios being less than a tenth of a ppt for most of the vertical column and modeled $HO_2$ mixing ratios in a range between 6 and 9 ppt.

### 3.1.5   Takeaways from uncertainties

Despite the geographic range of the regions presented here, there are many similarities to highlight. For instance, uncertainties

in GEOS-Chem modeled mixing ratios for ozone, OH, and $HO_2$ were largely similar. As a rule of thumb, uncertainties in ozone mixing ratios were around 20 %, OH between 25 and 40 %, and $HO_2$ between 20 and 35 %. Also, for most regions, when uncertainties in both GEOS-Chem and measurements are taken into account, there is general agreement between oxidant mixing ratios with the exception of ozone profiles in the higher altitude Houston based INTEX-B flights and ozone in a few other vertical bins in the Pacific INTEX-B flights.

## 3.2   Sensitivities

To explore from where the model-measurement disagreements may be coming, Figs. 6, 7, 8, and 9 show the median first order sensitivity indices across INTEX-A and regional INTEX-B flights for ozone, OH, and $HO_2$. As the sensitivities of ozone, OH and $HO_2$ varied with altitude, we show the analysis for the 0-1 km, 3-4 km, and 7-8 km vertical bins.

### 3.2.1   INTEX-A

Generally ozone was most sensitive to emissions, particularly $NO_x$ and isoprene (Fig. 6). Near the surface, ozone was most sensitive to the EPA-NEI (Environmental Protection Agency–National Emissions Inventory) $NO_x$ emissions and isoprene ($S_i$= 0.21 and 0.20 respectively). A few kilometers up, this sensitivity to surface $NO_x$ emissions is replaced by sensitivity to lightning $NO_x$ ($S_i$= 0.28 and 0.30 for 3-4 km and 7-8 km respectively). Sensitivity to chemical factors such as the $NO_2$ + OH reaction rate, and the $NO_2$ photolysis rate were largely altitude independent ($S_i$ between 0.08 and 0.13 for k[$NO_2$ + OH]; $S_i$= 0.07-0.08

for j[$NO_2$]).





Sensitivities for OH largely mirrored those of ozone (Fig. 6). As photolysis of ozone in the presence of water vapor leads directly to the production of OH, this is unsurprising. In addition to $NO_x$ and isoprene emissions mentioned with ozone, we also find OH above 3 km to be sensitive to CO emissions, especially from biomass burning ($S_i$= 0.16 between 3-4 km and $S_i$= 0.10 between 7-8km).

Near the surface where modeled aerosol concentrations are greatest, $HO_2$ is most sensitive to the aerosol uptake of $HO_2$ and isoprene emissions ($S_i$= 0.28 and 0.25 respectively) (Fig. 2). This sensitivity to aerosol uptake is reduced higher in the troposphere with biomass CO ($S_i$= 0.26 at 3-4 km and $S_i$= 0.18 between 7-8 km), lightning $NO_x$ ($S_i$= 0.12 at 7-8 km), and isoprene emissions ($S_i$= 0.15 between 3 and 4 km and $S_i$= 0.26 between 7 and 8 km) being the dominant sources of the uncertainty above 3 km. As uncertainty in gamma $HO_2$ is not limited to just the rate of the reaction, but also to the

product, we examined the modeled profiles in a model run having gamma $HO_2$ producing $H_2O_2$ rather than $H_2O$. With small differences generally around or less than half a ppt for $HO_2$ and likewise small differences for OH and ozone, $HO_2$ and the other oxidants are rather insensitive to this difference. Sensitivity to isoprene emissions is roughly altitude independent. As isoprene's lifetime is shorter than the timescales to allow consequential transport past the boundary layer, the sensitivity of $HO_2$ to isoprene emissions in the mid to free troposphere is almost certainly due to chemistry relating to secondary and higher

order isoprene products such as the photolysis of formaldehyde and acetaldehyde.

### 3.2.2  INTEX-B Houston

As with INTEX-A, ozone is largely sensitive to $NO_x$ emission inventories, specifically soil $NO_x$ near the surface and lightning $NO_x$ from 3 km higher (Fig. 7). In contrast to the height dependencies in the emissions inventories sensitivities, sensitivity to chemical factors were generally altitude independent with sensitivities to k[$NO_2$ + OH] ranging between $S_i$ values of 0.07 and

0.09 and j[$NO_2$] and j[$O_3$] between 0.03 and 0.08. For emission factors, in the lowest 1 km apart from soil $NO_x$ emissions ($S_i$= 0.28), we also find isoprene emissions ($S_i$= 0.08), and EDGAR $NO_x$ emissions ($S_i$= 0.07) having $S_i$ values greater than 0.05. From 3-4 km higher, lightning $NO_x$ becomes the dominant source of uncertainty with $S_i$ values of 0.30 around 4 km and higher between 7 and 8 km ($S_i$= 0.41). In these higher altitude bins, we also find ozone to have greater sensitivity to biomass CO emissions with $S_i$ values of 0.07 between 3 and 4 km and $S_i$= 0.09 between 7 and 8 km.

Similar to ozone, while we find OH to be most sensitive to emissions sources, the sensitivity to these sources are altitude dependent (Fig. 7). Near the surface, OH is most sensitive to isoprene and soil $NO_x$ emissions sources ($S_i$ values of 0.21 and 0.15 respectively). Chemical factors such as k[$NO_2$ + OH], aerosol uptake of $HO_2$, and j[$NO_2$] also had $S_i$ values greater than 0.05 (0.09, 0.08, and 0.07 respectively). Higher, lightning $NO_x$ becomes the dominant source of uncertainty for OH mixing ratios with $S_i$ values of 0.21 in the 3-4 km bin and 0.54 for the 7-8 km bin.

For $HO_2$ mixing ratios, near the surface we find gamma $HO_2$ to be responsible for about half of the model uncertainty ($S_i$= 0.51) with isoprene emissions being the only other factor with $S_i$ > 0.05 ($S_i$= 0.16) (Fig. 7). This dominance by gamma $HO_2$, though, is restricted to near the surface where aerosol concentrations are highest. In fact, higher than 3 km, we find biomass CO emissions to become the dominant source of uncertainty ($S_i$= 0.27 for 3-4 km, $S_i$= 0.38 for 7-8 km). Sensitivity to isoprene emissions is similar between 3-4 km and 7-8 km with $S_i$ values of 0.13 and 0.14 respectively.



### 3.2.3 INTEX-B Honolulu

For the flights based out of Honolulu, near surface ozone was most sensitive to surface emissions sources in the first vertical kilometer with ship $NO_x$ ($S_i$= 0.27) and methyl bromoform emissions ($S_i$= 0.07) and a variety of chemical factors such as the ozone photolysis rate (j[$O_3$] ($S_i$= 0.14), k[$NO_2$+oh] ($S_i$= 0.06), j[HOBr] ($S_i$= 0.05), j[$NO_2$] ($S_i$= 0.05)) (Fig. 8). Higher, ozone becomes sensitive to other emissions sources, especially lightning $NO_x$ ($S_i$= 0.11 and 0.25 at 3-4 km and 7-8 km respectively), and to a lesser extent, soil, and E. Asian $NO_x$ and isoprene emissions. These latter emissions sources are noteworthy as they illustrate the sensitivity of this region to non-local upwind emission sources as there are not any appreciable isoprene or soil $NO_x$ emissions over the remote north central Pacific. In addition to emissions sources, ozone also showed moderate sensitivity to chemical factors. In particular, the photolysis rate of ozone, in spite of its low uncertainty (20 %), had sensitivity indices ranging between 0.10 and 0.15 between the surface and 5 km. The $NO_2$ + OH reaction rate also had sensitivity indices about 0.07 at most altitudes.

OH mixing ratios were largely sensitive to the same factors as ozone (Fig. 8). Near the surface OH was largely sensitive to ship $NO_x$ emissions ($S_i$= 0.38), both biomass and E. Asia CO, j[$O_3$], k[$NO_2$ + OH], and j[$NO_2$] ($S_i$= 0.09, 0.08, 0.08, 0.06, and 0.05 respectively). Above 3 km, there is not any one factor that overwhelmingly contributes to the uncertainty, but CO and $NO_x$ emissions, along with the photolysis rate of ozone and the $NO_2$ + OH reaction rate all had $S_i$ values around greater than 0.05 for the higher altitude bins.

Like the Houston flights, $HO_2$ mixing ratios were largely sensitive to CO emissions, $NO_x$ emissions, and aerosol uptake of $HO_2$, only sensitivity to aerosol uptake is reversed vertically with higher sensitivities coming in the upper troposphere rather than near the surface ($S_i$= 0.10, 0.16, 0.30 for 0-1 km, 3-4 km, and 7-8 km vertical bins) (Fig. 8). This is a result of the modeled aerosol concentrations being highest near the surface for the Houston flights, and highest in the upper reaches of the troposphere for the Honolulu flights.

### 3.2.4 INTEX-B Anchorage

Near the surface, ozone sensitivity was dominated by ship $NO_x$ emissions ($S_i$= 0.52), and to a much lesser extent photolysis of HOBr ($S_i$= 0.06). Higher, a host of emissions factors become more important with bromoform emissions ($S_i$= 0.11 for 3-4 km and $S_i$= 0.09 for 7-8 km), soil $NO_x$ ($S_i$= 0.10 and 0.11 for 3-4 km and 7-8 km respectively), and lightning $NO_x$ ($S_i$= 0.13 at 7-8 km) (Fig. 9). Chemical factors such as k[$NO_2$ + OH] and j[$NO_2$] also were responsible for between 6 and 8 % of the uncertainty for both the 3-4 km and 7-8 km altitude bins.

Like ozone, OH was overwhelmingly sensitive to ship $NO_x$ emissions ($S_i$= 0.50) with this one factor being responsible for around half the model uncertainty (Fig. 9). At 3-4 km, this sensitivity to ship $NO_x$ emissions is replaced by CO emissions from E. Asia and biomass burning and soil $NO_x$ ($S_i$= 0.11 for E. Asia CO, $S_i$= 0.09 for biomass CO and soil $NO_x$). From 3 km higher, OH mixing ratios are most sensitive to the aerosol uptake of $HO_2$ ($S_i$= 0.14 at 3-4 km, $S_i$= 0.29 at 7-8 km).

At all but the highest altitudes, modeled $HO_2$ mixing ratios were overwhelmingly sensitive to the aerosol uptake of $HO_2$ (gamma $HO_2$) with this one factor contributing around half the model uncertainty ($S_i$= 0.49 at 0-1 km, $S_i$= 0.57 at both 3-4 km





and 7-8 km) (Fig. 9). This dominance of gamma $HO_2$ on $HO_2$ mixing ratios has been noted before in the similar ARCTAS-A (Arctic Research of the Composition of the Troposphere from Aircraft and Satellites) domain (Christian et al., 2017).

### 3.3 Discussion of results

Broadly speaking, measured and GEOS-Chem modeled oxidant profiles agreed to some extent in most of the cases outlined here. However, with 512 model runs for each campaign representing various combinations of perturbations to the inputs, it raises the question: which ensemble members fit the measured profiles best? With 512 model runs with various perturbations of the inputs, some members did come much closer to matching the measured profiles. In the following subsections we describe the commonalities among these better performing ensemble members' perturbations to $NO_x$ emissions and aerosol uptake.

### 3.3.1 $NO_x$ emissions

For all the regions presented here, GEOS-Chem modeled and measured ozone and OH profiles have closer agreement with lower lightning $NO_x$ emissions than emitted by default. In examining the closest 25 model ensemble members for each region and oxidant, we find reductions in their lightning $NO_x$ emissions anywhere from $\sim$25 % for Anchorage INTEX-B ozone and OH, INTEX-A ozone, Honolulu INTEX-B OH, to around a factor of 2 reduction for INTEX-A OH, Houston INTEX-B ozone and OH, and Honolulu INTEX-B ozone. Considering GEOS-Chem tended to overpredict ozone and OH, especially at higher altitudes, it is unsurprising there is better agreement with lower lightning $NO_x$ emissions.

The vertical profiles of NO and $NO_2$ (Fig. S1) somewhat corroborate this overestimate of $NO_x$ emissions in INTEX-A and can explain the overestimate of ozone. In INTEX-A, we found modeled $NO_2$ to be consistently greater than their respective measured values. Near the surface, this difference can be anywhere between 50 % and factor of 2 or greater for $NO_2$ with the greatest difference on an absolute basis near the surface (0-1 km) and on a percentage basis in the middle troposphere (between 5 and 7 km). In contrast to INTEX-A $NO_2$ mixing ratios, NO was underpredicted by the model with the exception of the first vertical kilometer. With high $NO_2$ and low NO, the model steady-state ozone concentrations would be elevated as ozone concentrations are generally proportional to the $[NO_2]/[NO]$ ratio (e.g., Chameides and Walker, 1973). In the Houston based INTEX-B flights, we found $NO_2$ to have modeled mixing ratios greater than measured between the surface and 1 km and above 5 km (Fig. S2). Between 5 and 9 km, NO and $NO_2$ mixing ratios are between 10 and 25 ppt too high in the model compared to measurements.

This model $NO_x$ overestimate is similar to results found in Travis et al. (2016) for the SEAC[4]RS campaign. In the case of Travis et al. (2016), GEOS-Chem more closely matched observations when the United States regional $NO_x$ emissions were reduced by a factor of 2. The blue lines in Figs. 10 and 11 illustrate the better model-measurement agreement, especially for ozone, when both EPA-NEI and lightning $NO_x$ emissions are reduced by a factor of 2 for INTEX-A and Houston based INTEX-B flights. In the case of lightning $NO_x$, this factor of 2 reduction is similar to the difference between modeled lightning $NO_x$ production in the tropics versus the midlatitudes (north of 23°N for North America).

In the case of the INTEX-A flights, this reduction in $NO_x$ emissions eliminates much of the model-measurement disagreement, especially for ozone, but unlike INTEX-A, the Houston based INTEX-B GEOS-Chem model-measurement disagreement





is not fully bridged for ozone, especially in the upper troposphere. This persistent disagreement suggests that lightning $NO_x$ emissions are not solely to blame for the upper altitude disagreement in ozone mixing ratios for the Houston based INTEX-B flights.

In addition to lightning $NO_x$, the Pacific flights of INTEX-B were also sensitive to ship $NO_x$ emissions, especially for the near surface vertical bins. For ozone, the 25 best matching model ensemble members had higher ship $NO_x$ emissions (65 % greater for Honolulu and 25 % greater for Anchorage flights). Since ozone was underpredicted by the model in conjunction with $NO_x$ (Figs. S3 and S4), increasing $NO_x$ emissions would presumably ameliorate some of this model-measurement disagreement. While this strong sensitivity to shipping emissions was not found during the ARCTAS campaign, this difference is likely a result of the more southerly direction, and thus more maritime domain, of the INTEX-B flights out of Anchorage, rather than the more continental flights of the ARCTAS campaign. Model treatment of ship emissions is unique in comparison to other anthropogenic sources. In order to approximate the complex and non-linear chemistry within ship exhaust plumes, $NO_x$ emissions are modified and partitioned via the PARAmeterization of emitted NOX (PARANOX) scheme into not only $NO_x$ emissions but also directly as ozone (Vinken et al., 2011). Clearly both the ship emissions and their immediate treatment is an important consideration, especially for near surface ozone and OH over remote maritime domains such as the Northern Pacific Ocean.

Underprediction of ozone and $HO_x$ is a persistent problem in this northern domain and largely mirrors previously published studies involving the ARCTAS campaign, a field campaign that took place over the North American Arctic in April of 2008 (Jacob et al., 2010; Alvarado et al., 2010). For the same flights, we similarly find model underprediction of $NO_x$ mixing ratios, especially above 2 km (Fig. S4). Underprediction of $NO_x$ mixing ratios would explain some of the underprediction of ozone mixing ratios.

### 3.3.2 Aerosol uptake

As for the aerosol uptake of $HO_2$, the sensitivity of $HO_2$ mixing ratios to this factor has been noted before (Martin et al., 2003; Mao et al., 2010; Christian et al., 2017), but mostly in the Arctic where low $NO_x$ mixing ratios and lower temperatures lead to longer $HO_2$ lifetimes. Indeed, we found greater sensitivity to this factor in the Anchorage based INTEX-B flights, the northernmost domain analyzed here. However, we also find similar sensitivities for $HO_2$ mixing ratios in different vertical bins for the other regions presented here. Like a similar study for a North American Arctic campaign (Christian et al., 2017), we also consistently find better agreement between $HO_2$ modeled and measured mixing ratios when aerosol uptake of $HO_2$ rates are reduced from its default rate of 0.20. In the case of the best 25 fitting ensemble member profiles, we find rates of anywhere between, 0.133 in Honolulu INTEX-B, 0.085 for Houston INTEX-B, 0.069 for INTEX-A, and 0.064 for Anchorage INTEX-B. For most of these cases, where we found greatest sensitivity to gamma $HO_2$, we also found $HO_2$ underprediction by GEOS-Chem. Thus, lower uptake rates alleviate some of this difference.

It is also possible that some of the underprediction of $HO_2$ by the model could be attributed to missing $HO_2$ sources or interferences in the measurements from peroxy radicals (Fuchs et al., 2011). As this interference requires the presence of alkenes or aromatics, it is more of a consideration near the surface and VOC emissions sources. While this is a consideration



for the near surface HO$_2$ model underestimate in INTEX-A, it is not a major consideration for INTEX-B since much of that campaign took place over more remote maritime regions.

## 4  Conclusions

We have presented a global sensitivity analysis of GEOS-Chem modeled oxidants for the time period and flight tracks of the INTEX-NA field campaigns. Uncertainties and sensitivities of modeled ozone, OH, and HO$_2$ were found and shown in Figs. 6, 7, 8, and 9. While there remains some disagreement between modeled and measured oxidant mixing ratios (Figs. 2, 3, 4, and 5), these differences were generally within the combined uncertainty ranges of both the modeled and measured values. In agreement with Travis et al. (2016), we find better model-measurement agreement for ozone with lower USA EPA-NEI emissions. With modeled ozone mixing ratios being most sensitive to lightning NO$_x$ in the middle and upper troposphere, we find similarly better model-measurement agreement with lower lightning NO$_x$ emissions for both INTEX-A and the INTEX-B Houston flights (Figs. 10 and 11). Recent work with parameterizing the nonlinear chemistry within lightning plumes in GEOS-Chem has found summertime Northern Hemispheric ozone and NO$_x$ concentrations to decrease (Gressent et al., 2016) so it is possible that improving the parameterization of lightning NO$_x$ may remedy some of this disagreement in future GEOS-Chem versions.

For some locations and altitudes, aerosol particle uptake of HO$_2$ can be responsible a large portion of uncertainty in HO$_2$ mixing ratios. In the case of the Anchorage based INTEX-B flights, gamma HO$_2$ was solely responsible for around half the uncertainty in HO$_2$ mixing ratios. While this sensitivity is not unexpected considering aerosol uptake of HO$_2$ has been shown to be important in poleward regions (Martin et al., 2003; Mao et al., 2010; Whalley et al., 2015; Christian et al., 2017), we also find considerable sensitivity to this factor in more southerly locations as well (Figs. 6, 7, 8, and 9). Similar to previous work for the ARCTAS campaign, we also find in all the regions presented here that lower uptake rates produce better model-measurement agreement (between 0.06 and 0.13 depending on the region as opposed to the standard 0.20). With varied locations showing sensitivity to gamma HO$_2$, it appears that in order to model HO$_2$ with accuracy and certainty, aerosol uptake needs to be well accounted for and understood.

While the sensitivity results were different depending on the domain, the picture is similar from a distance. Emissions tended to be the dominant source of uncertainty for the modeled oxidants presented here, even for remote maritime domains. In all the cases, near surface ozone and OH are most sensitive to surface emissions sources, especially NO$_x$ and, to a lesser extent, isoprene. We find similar sensitivities to lightning NO$_x$ above 3 kilometers. For HO$_2$, carbon monoxide emissions, especially from biomass burning, and isoprene emissions are the dominant emissions uncertainty sources. Despite their considerably lower uncertainty, chemical factors such as kinetic rate coefficients, especially the NO$_2$ + OH reaction rate, and photolysis rates, such as those of ozone and NO$_2$ also were responsible for a considerable portion of the uncertainty. This is noteworthy considering uncertainties in these chemical factors tend to be much lower than those for emissions sources ($\sim$20-30 % vs. factors of 2-3 for emissions). This highlights the value in not only reducing emissions uncertainties, but also in making more laboratory measurements to provide more certainty for chemical factors, even those thought to be well known.



*Acknowledgements.* We would like to acknowledge NASA's Atmospheric Composition Campaign Data Analysis and Modeling program (ACCDAM) for funding this project (grant NNX14AP43G), University Maryland's Cooperative Institute for Climate and Satellites (funded under a NOAA Cooperative Agreement), Harvard University for managing and supporting GEOS-Chem, GEOS-Chem support for assistance, Melody Avery for ozone measurements, and the rest of the INTEX science team for other aircraft measurements.





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





**Table 1.** Factors included in INTEX-A RS-HDMR analysis and their respective uncertainties. OC is organic carbon, MP is methylhydroperoxide, and $MO_2$ is methylperoxy radical. Uncertainties are expressed as multiplicative factors, except as noted in meteorological factors.

| Factor | Uncertainty[#] | Factor | Uncertainty[#] |
|---|---|---|---|
| **Emissions** | | **Photolysis** | |
| Biomass CO, $NO_x$, OC | | j [$BrNO_3$] | 1.4[d] |
| Soil $NO_x$ | 3.0[a] | j [$CH_2O$] | 1.4[d] |
| Methyl Bromoform ($CHBr_3$) | | j [$H_2O_2$] | 1.3[d] |
| EPA (USA) CO, $NH_3$, $NO_x$ | | j [$HNO_3$] | 1.3[d] |
| Streets (E. Asian) CO, $NH_3$, $NO_x$, $SO_2$ | 2.0 | j [HOBr] | 2.0[d] |
| Ship $NO_x$ | | j [$NO_2$] | 1.2[d] |
| Isoprene | 2.0[b] | j [$O_3$] | 1.2[d] |
| Lightning $NO_x$ | 2.0[c] | **Meteorology** | |
| **Kinetics** | | Cloud mass flux | 1.5[f] |
| k [$HNO_2$] [OH] | 1.5[d] | Relative humidity | 5 %[g] |
| k [$HNO_3$] [OH] | 1.2[d] | Soil Wetness | 8.8 %[e] |
| k [$HO_2$] [$HO_2$] | 1.15 / 1.2[*d] | Specific Humidity | 5 %[g] |
| k [$HO_2$] [NO] | 1.15[d] | Temperature | 1.8 K[e] |
| k [$MO_2$] [$HO_2$] | 1.3[d] | **Heterogeneous** | |
| k [$MO_2$] [NO] | 1.15[d] | Gamma $HO_2$ | 3.0[d] |
| k [$NO_2$] [OH] | 1.3[d] | | |
| k [$O_3$] [$HO_2$] | 1.15[d] | | |
| k [$O_3$] [NO] | 1.1[d] | | |
| k [OH] [$CH_4$] | 1.1[d] | | |
| k [OH] [$HO_2$] | 1.15[d] | | |

[#] at $1\sigma$ uncertainty confidence; [*] high pressure limit / low pressure limit uncertainties; [a] Jaeglé et al. (2005); [b] Guenther et al. (2012); [c] Liaskos et al. (2015); [d] Sander et al. (2011); [e] GEOS5-GEOS4; [f] Ott et al. (2009); [g] Heald et al. (2010)




**Table 2.** Factors included in INTEX-B RS-HDMR analysis and their respective uncertainties. OC is organic carbon, MP is methylhydroper-oxide, and $MO_2$ is methylperoxy radical. Uncertainties are expressed as multiplicative factors, except as noted in meteorological factors.

| Factor | Uncertainty[#] | Factor | Uncertainty[#] |
|---|---|---|---|
| **Emissions** | | **Photolysis** | |
| Biomass CO, $NH_3$, $NO_x$, OC | | j [$CH_2O$] | $1.4^d$ |
| Soil $NO_x$ | $3.0^a$ | j [$H_2O_2$] | $1.3^d$ |
| Methyl Bromoform ($CHBr_3$) | | j [$HNO_3$] | $1.3^d$ |
| EDGAR $NO_x$ | | j [HOBr] | $2.0^d$ |
| EMEP (European) $NO_x$ | | j [MP] | $1.5^d$ |
| EPA (USA) CO, $NO_x$ | 2.0 | j [$NO_2$] | $1.2^d$ |
| Streets (E. Asian) CO, $NH_3$, $NO_x$, $SO_2$ | | j [$O_3$] | $1.2^d$ |
| Ship $NO_x$ | | **Meteorology** | |
| Strat-Trop Exchange $O_3$ | | Cloud fraction | $8.5\ \%^e$ |
| Isoprene | $2.0^b$ | Cloud mass flux | $1.5^f$ |
| Lightning $NO_x$ | $2.0^c$ | Relative Humidity | $5\ \%^g$ |
| **Kinetics** | | Soil Wetness | $8.8\ \%^e$ |
| k [$HNO_3$] [OH] | $1.2^d$ | Specific Humidity | $5\ \%^g$ |
| k [$HO_2$] [$HO_2$] | $1.15 / 1.2^{*d}$ | Temperature | $1.8\ K^e$ |
| k [$HO_2$] [NO] | $1.15^d$ | U Wind | $0.71\ ms^{-1e}$ |
| k [$MO_2$] [$HO_2$] | $1.3^d$ | **Heterogeneous** | |
| k [$MO_2$] [NO] | $1.15^d$ | Gamma $HO_2$ | $3.0^d$ |
| k [MP] [OH] | $1.4^d$ | Gamma $NO_2$ | $3.0^d$ |
| k [$NO_2$] [OH] | $1.3^d$ | Henry's Law HOBr | $10.0^d$ |
| k [$O_3$] [$HO_2$] | $1.15^d$ | | |
| k [$O_3$] [NO] | $1.1^d$ | | |
| k [$O_3$] [$NO_2$] | $1.15^d$ | | |
| k [OH] [$CH_4$] | $1.1^d$ | | |
| k [OH] [$HO_2$] | $1.15^d$ | | |

# at $1\sigma$ uncertainty confidence; *high pressure limit / low pressure limit uncertainties; [a]Jaeglé et al. (2005); [b]Guenther et al. (2012); [c]Liaskos et al. (2015); [d]Sander et al. (2011); [e]GEOS5-GEOS4; [f]Ott et al. (2009); [g]Heald et al. (2010)





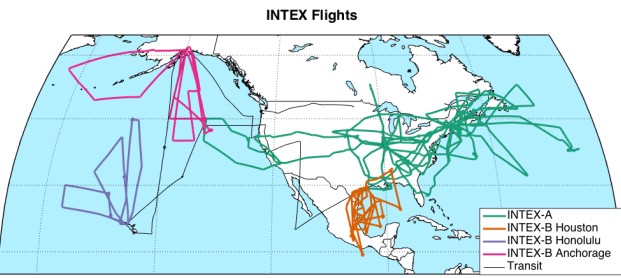

**Figure 1.** Map of INTEX-A & INTEX-B flights

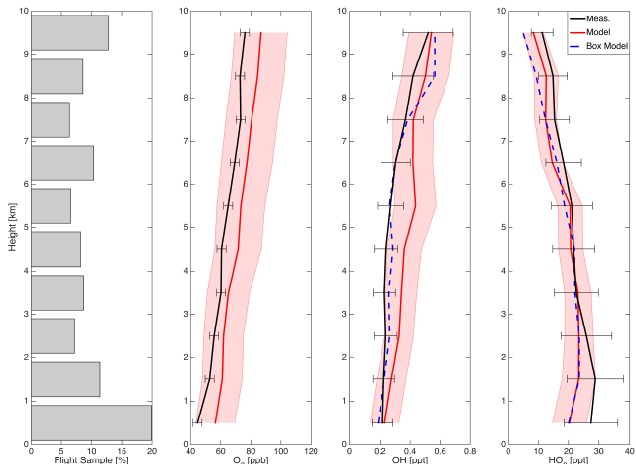

**Figure 2.** Vertical profiles of median modeled (red) and measured (black) ozone, OH, and $HO_2$ for INTEX-A flight data binned by kilometer. Gray bar graph shows percent of flight data within each vertical bin. Shaded regions represent $1\sigma$ of model ensemble; error bars on measurements are uncertainty at $1\sigma$ confidence. Blue line represents results from box model (Ren et al., 2008).





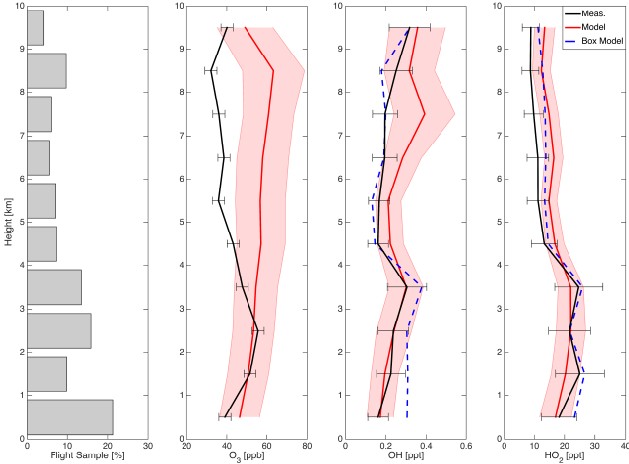

**Figure 3.** Vertical profiles of median modeled (red) and measured (black) ozone, OH, and $HO_2$ for Houston based INTEX-B flight data binned by kilometer. Gray bar graph shows percent of flight data within each vertical bin. Shaded regions represent $1\sigma$ of model ensemble; error bars on measurements are uncertainty at $1\sigma$ confidence. Blue line represents results from box model

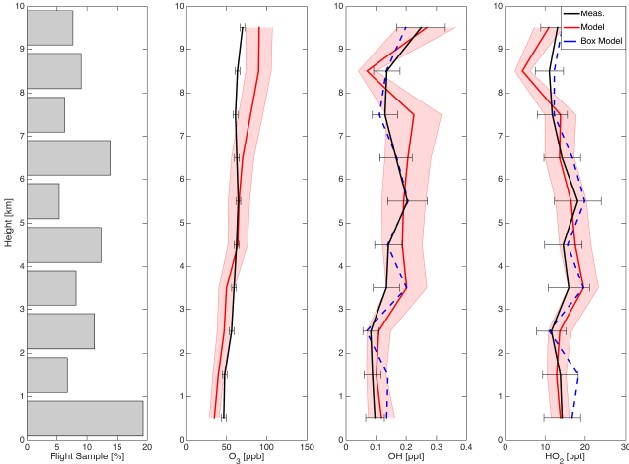

**Figure 4.** Vertical profiles of median modeled (red) and measured (black) ozone, OH, and $HO_2$ for Honolulu based INTEX-B flight data binned by kilometer. Gray bar graph shows percent of flight data within each vertical bin. Shaded regions represent $1\sigma$ of model ensemble; error bars on measurements are uncertainty at $1\sigma$ confidence. Blue line represents results from box model





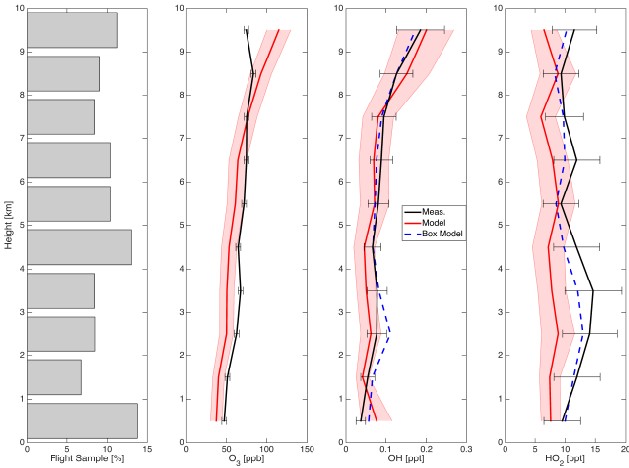

**Figure 5.** Vertical profiles of median modeled (red) and measured (black) ozone, OH, and HO$_2$ for Anchorage based INTEX-B flight data binned by kilometer. Gray bar graph shows percent of flight data within each vertical bin. Shaded regions represent 1$\sigma$ of model ensemble; error bars on measurements are uncertainty at 1$\sigma$ confidence. Blue line represents results from box model

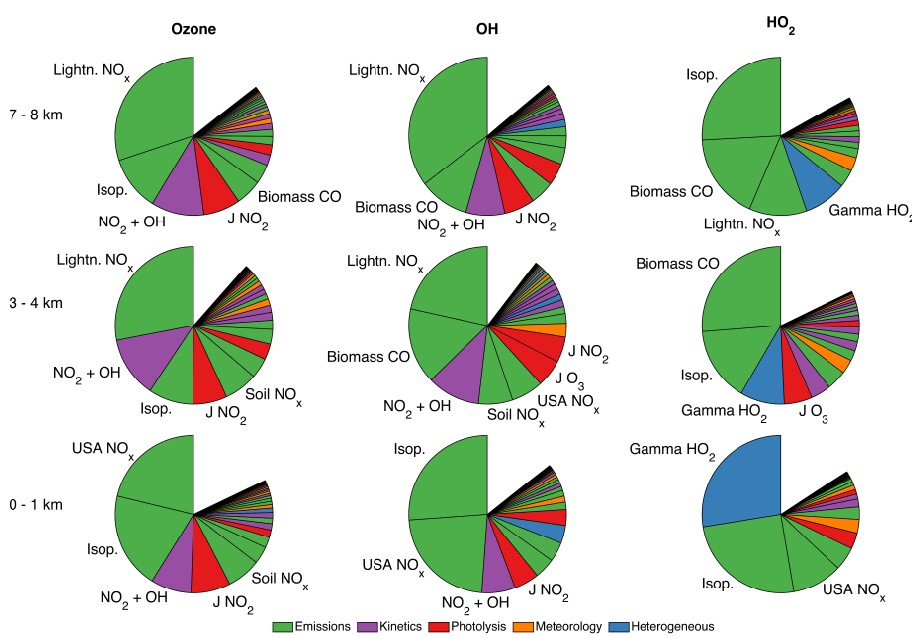

**Figure 6.** First order sensitivity indices for median flight track ozone, OH, and HO$_2$ for INTEX-A flights. Legend categories are defined in Table 1.





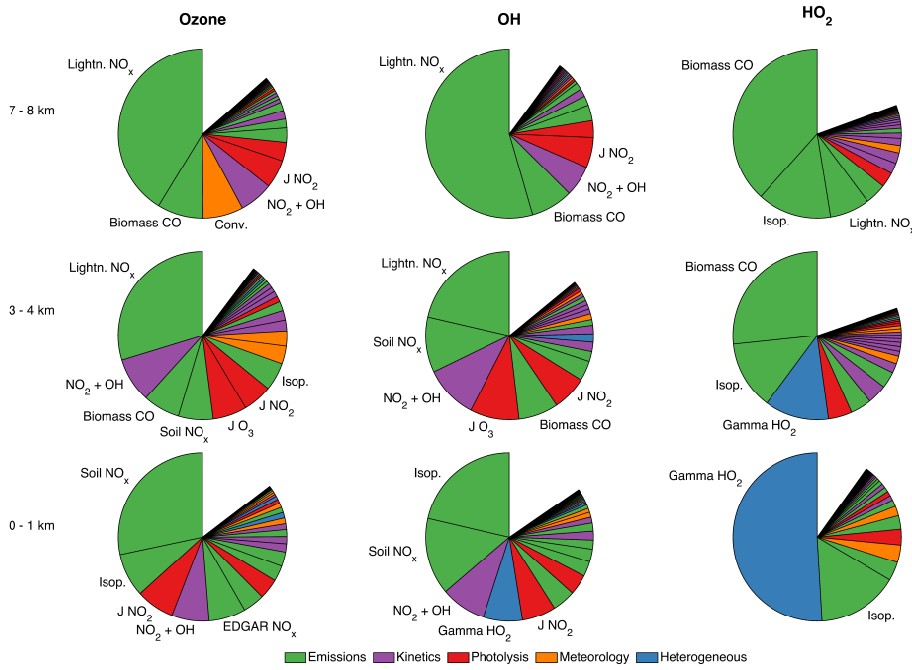

**Figure 7.** First order sensitivity indices for median flight track ozone, OH, and HO$_2$ for INTEX-B flights originating from and terminating in Houston. Legend categories are defined in Table 2.

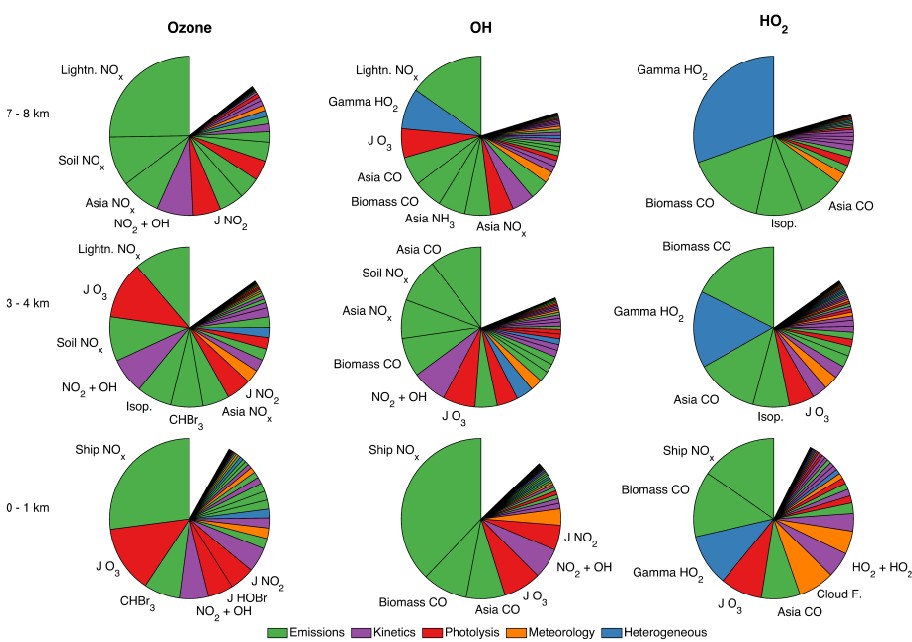

**Figure 8.** First order sensitivity indices for median flight track ozone, OH, and HO$_2$ for INTEX-B flights originating from and terminating in Honolulu. Legend categories are defined in Table 2.



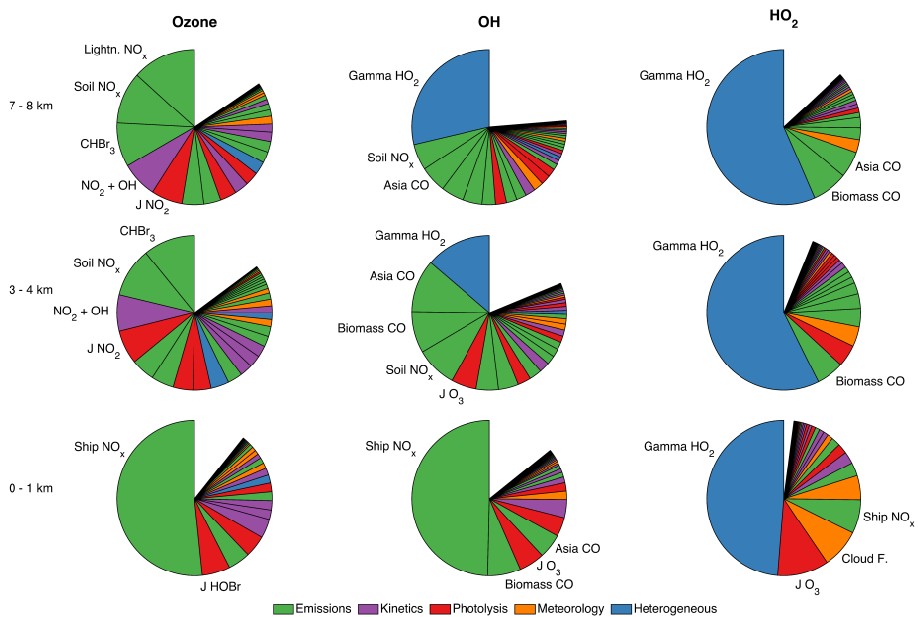

**Figure 9.** First order sensitivity indices for median flight track ozone, OH, and $HO_2$ for INTEX-B flights originating from and terminating in Anchorage. Legend categories are defined in Table 2.

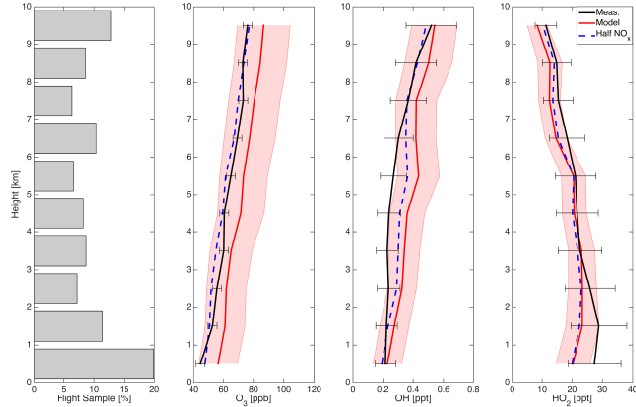

**Figure 10.** Vertical profiles of median modeled (red) and measured (black) ozone, OH, and $HO_2$ INTEX-A flight data binned by kilometer. Gray bar graph shows percent of flight data within each vertical bin. Shaded regions represent $1\sigma$ of model ensemble; error bars on measurements are uncertainty at $1\sigma$ confidence. Blue line represents EPA-NEI and lightning $NO_x$ emissions reduced by 50 %





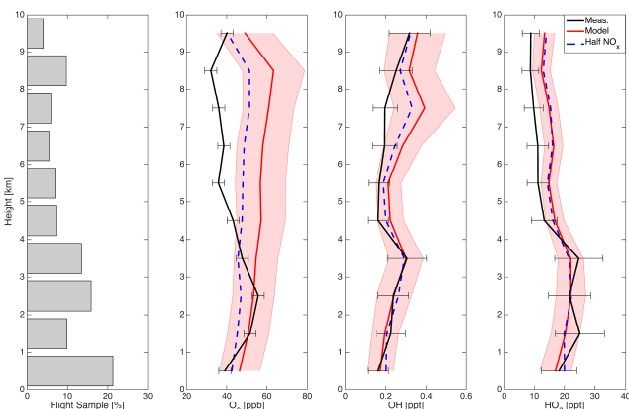

**Figure 11.** Vertical profiles of median modeled (red) and measured (black) ozone, OH, and HO$_2$ Houston based INTEX-B flight data binned by kilometer. Gray bar graph shows percent of flight data within each vertical bin. Shaded regions represent 1$\sigma$ of model ensemble; error bars on measurements are uncertainty at 1$\sigma$ confidence. Blue line represents EPA-NEI and lightning NO$_x$ emissions reduced by 50 %