# Peer review of "Global sensitivity analysis of GEOS-Chem modeled ozone and hydrogen oxides during the INTEX campaigns"

_Atmospheric Chemistry and Physics, 2017_

## Referee Comment (RC1) · P. Kasibhatla (Referee) · 25 Sep 2017

The focus of this paper is on an analysis of the causes of discrepancies between modeled and measured O3, OH, and HO2 vertical profiles during the INTEX-A and INTEX-B field campaigns. The analysis is based on a global sensitivity analysis approach, in which an ensemble of model runs (in which multiple variables are simultaneously perturbed) is used to construct sensitivity factors to delineate the relative importance of the various variables considered on modeled tracer fields. This is potentially an interesting approach to understand observation-model differences, but the paper seems to fall short in fully exploiting the power of this approach and in terms of the analysis

presented. I discuss my specific concerns below:

1) Section 2.2 presents a relatively technical description of the global sensitivity analysis approach and gives the impression that the advantage of the approach (relative to a local sensitivity analysis) is to examine the uncertainty in model results due to the joint uncertainty in multiple model inputs. However, the paper focuses solely on the calculation and analysis of first order sensitivity indices, because of the computational burden associated with number of model runs needed to estimate higher order sensitivity indices. This raises the question as to whether the calculated first order sensitivity indices are in fact meaningful, or whether they themselves could be uncertain owing to the truncation of the polynomial function (eqn 1) that is being fit. It also raises the question as whether there is any advantage of using the global sensitivity analysis approach itself. Given that only first order sensitivity indices are estimated, wouldn't it have been more straightforward to use a local sensitivity analysis approach?

2) Another potentially important issue pertains to the treatment of uncertainty of individual variables. Let me illustrate by focusing on the assumed uncertainty for biomass burning emissions. Presumably, the authors assume that this is a systematic (as opposed to random) uncertainty so that in any given model run, the sampled uncertainty factor is applied uniformly in each and every grid cell of the model. Is this in fact appropriate? Or would it more appropriate to assume that some portion of the uncertainty is random? Also unclear is how inter-species uncertainty correlations are handled. For example, are CO and NOx biomass burning emissions perturbed by the same scaling factor in every grid cell in a given run (which would occur if the uncertainty was solely due to uncertainty in burned area for example) or are the perturbation factors completely independent (which would occur if the uncertainty was solely due to uncertainty in emission factors for examples)? As another example, are the perturbation factors for the photolysis rates correlated or uncorrelated? The authors should describe more clearly their approach in selecting perturbation factors and the justification for the approach they use – and discuss how their choice might impact their conclusions.

[Figure]

3) There also seems to be a bit of a disconnect between the sensitivity indices shown in Figures 6-9, and the discussion of results in Section 3.3. For example, on page 13, line 1 the authors say that '. . .suggests that ..'. Why 'suggests'? Doesn't Figure 7 in fact make the case that uncertainty in lightning NOx cannot solely explain the discrepancy in modeled O3? More importantly, I am somewhat puzzled by the authors approach of using a subset of ensemble members to illustrate some of their points. Wouldn't it be more straightforward to make an additional set of model runs in which all the important identified parameters were appropriately perturbed (based on Figures 6-9) and to demonstrate that the the final configuration results in better statistical agreement with observations?

4) A minor comment - I think some thought needs to be put into making the presentation more appealing. Much of Section 3 describes in detail various aspects of the figures that are obvious by simply looking at the figures, rather than highlighting the most important aspects of the results.

---

## Referee Comment (RC2) · Anonymous Referee #2 · 28 Sep 2017

**1   General comments**

This manuscript presents a global uncertainty analysis of the concentration of oxidants ($O_3$, $OH$, and $HO_2$) at various altitudes through the troposphere, and in four geographical regions (central and northeastern U.S. and Canada, Gulf of Mexico, Pacific ocean near Honolulu, Hawaii, and Pacific ocean near the southern coast of Alaska). The authors use an ensemble of 512 GEOS-Chem simulations in which various inputs have been pseudo-randomly perturbed within prescribed uncertainty ranges, along with the high dimensional model representation (HDMR) technique to apportion the uncertainty in modeled oxidant concentrations to each of the perturbed inputs. The geographical

regions studied in the manuscript feature various chemical and meteorological regimes and different local and upwind emissions profiles. Comparison of the results for these regions brings valuable insight into the model inputs that influence oxidant concentrations in these various conditions. The study is well conducted and the results are clearly presented and explained.

A concern for the publication of this manuscript is the similarity of this study with a previous study (Christian *et al.*, 2017a), mostly by the same authors. I do think that the proposed manuscript brings significant new contributions that warrant publication, but the authors should discuss more explicitly the insights that are novel and significant in this manuscript compared to the authors' previous work. The sections below describe in more detail the suggestions and comments that I would ask the authors to address prior to publication.

**2  Specific comments**

- As mentioned in the Overview section, the authors should discuss the novel insights that this study brings compared to the previous work of Christian *et al.* (2017a). One novel and insightful aspect of the proposed manuscript seems to be the comparison of the uncertainty apportionment between different regions, as well as the vertical resolution of the analysis.

- A similarly worded description of HDMR is already present in the previous work. Although it is useful for the proposed manuscript to summarize the principal concepts of this method, I suggest this description be re-worded further. The same comment applies to other parts of the "Methods" section.

- Page 4, Equation (1): shouldn't $f_i(x_j)$ be $f_i(x_i)$ instead?

- Page 2, Lines 16-22: "Instead, the sensitivity analyses of GEOS-Chem modeled

results has either used local methods in which the factor of interest is perturbed individually and compared to the model state without this perturbation, or the GEOS-Chem adjoint (Henze et al., 2007). [...] While useful in determining some individual sensitivities, these methods neither can nor were intentioned to provide a complete picture of model sensitivities in which many inputs have uncertainties."

The adjoint sensitivity technique can be used to efficiently calculate first-order sensitivities of a model metric or cost function to many model inputs (sensitivities of a given model metric or cost function to all model inputs can be efficiently calculated with a single "adjoint simulation"). Although the results from such an analysis provide "only" first-order local sensitivities, one can argue that they do provide a fairly comprehensive picture of model sensitivities for a given metric and a large number of model inputs that have uncertainties.

A strength of the HDMR method used in the proposed manuscript resides in the fact that model non-linearities are accounted for in the propagation of uncertainties, while other sensitivity approaches are often limited to first-order sensitivities. However, the HDMR approach does require a large number of model simulations (512 here). Additionally, the apportionment of the overall uncertainty with the HDRM method relies on *a priori* estimates of the uncertainties on relevant inputs. Sensitivity or uncertainty apportionments based on other sensitivity methods often do not depend on such *a priori* estimates.

Could the authors discuss these considerations in greater detail in the manuscript prior to publication?

- Page 3, Lines 18–20: "As uncertainties are not published for the meteorological models, we define our meteorological uncertainties as the average of the monthly standard deviations of the difference between GEOS-4 and GEOS-5 meteorological fields for 2005, a year of overlap between the models."

How different are these two models? If they are fairly similar, the uncertainties

on meteorological inputs may be significantly underestimated. Can the authors discuss the fairness of this assumption?

- Page 3, Lines 12–13: "In general, there were typically small differences between modeled results using either $4° \times 5°$ or $2° \times 2.5°$ resolutions but we illustrate in our results where this is not the case."

  The authors do discuss some of these differences in the Results section (for example: section 3.1.2), but it would be insightful to see more quantitative information describing the model-versus-observations agreement (for example: mean bias, standard deviation) with the lower resolution simulations on the one-hand, compared to the higher resolution simulations on the other hand.

- Page 5, Equations (3) and (4): I am unsure as to whether $\varphi_r^i$ in equation (3) is the same as $\varphi_p^i$ and $\varphi_q^j$ in equation (4) if $r = p = q$ and $i = j$. Additionally, the use of superscripts as indices can also introduce confusion. Can the authors add text to clarify these concepts?

- Can the authors discuss the contributions of uncertainties associated with inputs interacting with one another (*i.e.* "missing" slices in Figures 6–9), and the significance of these missing slices for the interpretation of the results presented in the manuscript?

**3 Technical corrections**

- The authors repeatedly use the word "standard" to refer to notions such as "common practice" or "default value" or "default configuration". I suggest that the word "standard" be reserved for a more restrictive meaning of the word (*i.e.* a formalized norm or convention). Examples:

- Page 3, Line 8: "We use in this study the standard GEOS-Chem model"
- Page 3, Line 21: "the model ensemble made use of the standard emissions inventories"
- Page 4, Line 7: "the standard model treatment"
- Page 14, Line 21: "as opposed to the standard 0.20"

In what follows, text that I suggest be removed is written inside curly braces and in red, and suggested replacement text is in blue.

- Page 2, Line 15: "save for {a} some recent work"

- Page 3, Line 8: "We use in this study the standard GEOS-Chem model (v9-02), a {popular}widely-used global chemical transport model"

- Page 3, Line 21: "{For much of the developed world}For many industrialized regions"

- Page 6, line 29: "During INTEX-A, the NASA DC-8 primarily sampled the eastern half of the United States and Canada {INTEX-A} during the summer of 2004"

- Page 9, Line 5–6: "aerosol uptake {to}of $HO_2$"

- Page 9, Lines 8–9: "In contrast, {uncertainty}uncertainties in both $OH$ and $HO_2$ mixing ratios were considerable"

- Figures 6–9: the different colors used in these Figures translate to very similar shades of gray when converted to gray-scale. I suggest changing some of these colors so that the different categories of inputs (Emissions, Kinetics, Photolysis, Meteorology, Heterogeneous) can be more easily distinguished when these Figures are converted to gray-scale. I also suggest showing on these Figures the

numerical values corresponding to the slices (*i.e.* contribution of each input to total uncertainty, in %), at least for the largest slices.

- Acknowledgments: "University Maryland". Missing word "of"?

**4  Reference(s)**

Christian, K. E.; Brune, W. H.; Mao, J. 2017a. Global sensitivity analysis of the GEOS-Chem chemical transport model: ozone and hydrogen oxides during ARCTAS (2008). *Atmospheric Chemistry and Physics*, 17 (5) 3769–3784.

---

## Author Comment (AC1) · 14 Nov 2017

We thank Prof. Kasibhatla for his thorough review and thoughtful suggestions for improving the manuscript. Below are our responses to his comments (*italics*).

*The focus of this paper is on an analysis of the causes of discrepancies between modeled and measured $O_3$, OH, and $HO_2$ vertical profiles during the INTEX-A and INTEXB field campaigns. The analysis is based on a global sensitivity analysis approach, in which an ensemble of model runs (in which multiple variables are simultaneously perturbed) is used to construct sensitivity factors to delineate the*

*relative importance of the various variables considered on modeled tracer fields. This is potentially an interesting approach to understand observation-model differences, but the paper seems to fall short in fully exploiting the power of this approach and in terms of the analysis presented. I discuss my specific concerns below:*

*1. Section 2.2 presents a relatively technical description of the global sensitivity analysis approach and gives the impression that the advantage of the approach (relative to a local sensitivity analysis) is to examine the uncertainty in model results due to the joint uncertainty in multiple model inputs. However, the paper focuses solely on the calculation and analysis of first order sensitivity indices, because of the computational burden associated with number of model runs needed to estimate higher order sensitivity indices. This raises the question as to whether the calculated first order sensitivity indices are in fact meaningful, or whether they themselves could be uncertain owing to the truncation of the polynomial function (eqn 1) that is being fit. It also raises the question as to whether there is any advantage of using the global sensitivity analysis approach itself. Given that only first order sensitivity indices are estimated, wouldn't it have been more straightforward to use a local sensitivity analysis approach?*

Response: As for calculating the model sensitivity to all the various inputs, there would be small if any benefit to performing local sensitivity analyses. With ∼30-50 inputs, it would still take hundreds of model runs to create polynomials, or a sort of regression, relating the model inputs to the outputs. Not all the model inputs are described in simple linear functions, many of the component functions of the RS-HDMR analysis are $2^{nd}$ and higher ordered polynomials describing the model output response to perturbations in each individual model input. Also, through a series of local sensitivity analyses, we would lose somewhat the contextualization of the different sensitivities to the model inputs. It is true that we generally find non-linearities to contribute a rather small portion of the total model uncertainty. This means that modelers wishing to determine the sensitivity of these modeled oxidants to one individual factor can likely assume these

factor interactions are small to negligible compared to their separate effects. This point it made clearer in the conclusions (P14 L31-P15 L2). One of the strengths in this method is that we don't assume linearity between factors and factor-factor interactions are accounted for.

We have confidence in the model sensitivities calculated from testing the sensitivity of the sensitivity indices to varying the number of model runs included. For example, calculating the sensitivity indices using 512, 448, 384, 320, 256, 192, and 128 model runs. As noted on P4 L28-30, we find little difference in the sensitivity indices calculated. This insensitivity to increased number of model runs, especially from 256 higher gives us confidence in these first order sensitivity indices. This result is similar to Lu et al., 2013.

Changes: Added a couple sentences at the end of P14 (P14 L31-P15 L2) to make note of the generally small contributions by factor-factor interactions in the overall model uncertainty.

*2. Another potentially important issue pertains to the treatment of uncertainty of individual variables. Let me illustrate by focusing on the assumed uncertainty for biomass burning emissions. Presumably, the authors assume that this is a systematic (as opposed to random) uncertainty so that in any given model run, the sampled uncertainty factor is applied uniformly in each and every grid cell of the model. Is this in fact appropriate? Or would it more appropriate to assume that some portion of the uncertainty is random? Also unclear is how inter-species uncertainty correlations are handled. For example, are CO and $NO_x$ biomass burning emissions perturbed by the same scaling factor in every grid cell in a given run (which would occur if the uncertainty was solely due to uncertainty in burned area for example) or are the perturbation factors completely independent (which would occur if the uncertainty was solely due to uncertainty in emission factors for examples)? As another example, are the perturbation factors for the photolysis rates correlated or uncorrelated? The*

*authors should describe more clearly their approach in selecting perturbation factors and the justification for the approach they use - and discuss how their choice might impact their conclusions.*

Response: We treat all the perturbations for each factor independently from the others. There are a few reasons for this. The uncertainties we use for the emissions inventories largely come from the differences among different studies. Thus, this uncertainty is more closely a measure of systematic uncertainty in these emissions inventories. In the case of biomass burning emissions, as the reviewer notes, there are uncertainties in parts of this process that would affect all the emissions similarly (i.e., area burned, elapsed time of burn, etc.) but there are also uncertainties in other parts of this process that would not affect all emissions similarly (i.e., land cover, fuel type, temperature of burn). We felt with these uncertainties in the uncertainty it made more sense to treat all the emissions factors individually and separately allowing for us to determine the specific species responsible for model uncertainty. While the uncertainty in these biomass emissions are largely a function of processes that affect all the biomass emissions, like area and elapsed time of burn, there are still uncertainties in the partitioning of these emissions into various specific trace gases (Andreae and Merlet, 2001; van der Werf et al., 2010). Treating the emissions separately also allows for us to determine the specific emissions or processes that are resulting in model uncertainty. Lumping the uncertainty would lose some of these insights. It is not immediately obvious how perturbing some of these emissions factors in concert with one another would change the conclusions of this study. We already conclude that emissions factors form Asia, North America, and biomass burning are responsible for considerable portions of the total model uncertainty so grouping many of them together would presumably only serve to group their effects into a bigger piece of the pie.

The perturbations to photolysis rates are treated individually and systematically as well. These uncertainties come from the combined cross-sectional area and

quantum yield uncertainties in the JPL evaluation cited. As these uncertainties in the cross-sectional area and quantum yield pertain to the individual chemical species, we believe this is the appropriate way to express this given that errors in chemical rates would affect the global troposphere similarly worldwide. The inclusion of cloud fraction as a perturbed model input would contain some of this combined photolytic uncertainty (at least for INTEX-B).

Changes: Added a couple sentences (P4 L7-9) noting the independence of the perturbations and the short rationale for this.

*3. There also seems to be a bit of a disconnect between the sensitivity indices shown in Figures 6-9, and the discussion of results in Section 3.3. For example, on page 13, line 1 the authors say that '. . .suggests that ..'. Why 'suggests'? Doesn't Figure 7 in fact make the case that uncertainty in lightning $NO_x$ cannot solely explain the discrepancy in modeled $O_3$? More importantly, I am somewhat puzzled by the authors approach of using a subset of ensemble members to illustrate some of their points. Wouldn't it be more straightforward to make an additional set of model runs in which all the important identified parameters were appropriately perturbed (based on Figures 6-9) and to demonstrate that the the final configuration results in better statistical agreement with observations?*

Response: In our analysis we try to be restrained in our language and conclusions. The reviewer is correct in noting that any way we perturb the lightning $NO_x$ in the model fails to bridge the model-measurement gap, especially in the higher altitudes of the Houston INTEX-B flights (Figure 11). The reason we were not more confident with our language in this discussion is that there remains uncertainty in the way lightning $NO_x$ is parameterized and handled, especially in global models like GEOS-Chem. We note in our conclusions that there are some different lightning $NO_x$ parameterizations

and treatment in the works. Our tests only varied lightning $NO_x$ within the context of the existing parameterization which leaves the possibility that lightning $NO_x$, or specifically its parameterization, could be still be the culprit.

Also worth noting is that model sensitivity does not necessarily mean that the default treatment of that factor is "wrong". In some cases, we have found sensitivity to a factor and the best matching ensemble members had values close to the default model values. Just because a factor takes up a large piece of the pie charts in Figures 6-9, doesn't mean that that factor is "wrong".

As for the second half of the point raised, we do not see a great value in creating runs containing all the best matching perturbations and think that this could be ripe for mis-interpretation. The perturbations producing the best model-measurement agreement for one of the oxidant species studied do not necessarily produce the best model-measurement agreement in the other domains or among the other oxidants in the same domain. This lack of predictability limits the usefulness in creating these "improved" model runs. Rather the purpose of this section is to highlight processes that may be systematically misrepresented in the model (like gamma $HO_2$, lightning $NO_x$) and stimulate discussion for the others.

We included the model runs with lower $NO_x$ emissions due to the persistence of better model-measurement agreement with lower $NO_x$ emissions and to compare these results to those of the Travis et al. (2016) paper.

*4. A minor comment - I think some thought needs to be put into making the presentation more appealing. Much of Section 3 describes in detail various aspects of the figures that are obvious by simply looking at the figures, rather than highlighting the most important aspects of the results.*

We feel that we do highlight in these sections the important aspects of the results in

these sections. While it can be a bit repetitive in describing some results contained in the figures, we felt it was needed so the reader would know the model uncertainties in the profiles and specific sensitivity index values for the pie charts. Throughout the section we make note of the interesting/main picture elements and further expand upon these takeaways in the discussion of the results.

References:

Andreae, M. O., and Merlet, P.: Emission of trace gases and aerosols from biomass burning, Global Biogeochemical Cycles., 15(4), 955-966, doi:10.1029/2000GB001382, 2001.

van der Werf, G. R., Randerson, J. T., Giglio, L., Collatz, G. J., Mu, M., Kasibhatla, P. S., Morton, D. C., DeFries, R. S., Jin, Y., and van Leeuwen, T. T.: Global fire emissions and the contribution of deforestation, savanna, forest, agricultural, and peat fires (1997-2009), Atmos. Chem. Phys., 10, 11707-11735, https://doi.org/10.5194/acp-10-11707-2010, 2010.

Lu, X.,Wang, Y.-P., Ziehn, T., and Dai, Y.: An efficient method for global parameter sensitivity analysis and its applications to the Australian community land surface model (CABLE), Agricultural and Forest Meteorology, 182–183, 292–303, doi:10.1016/j.agrformet.2013.04.003, http://www.sciencedirect.com/science/article/pii/S0168192313000804, 2013.

Travis, K. R., Jacob, D. J., Fisher, J. A., Kim, P. S., Marais, E. A., Zhu, L., Yu, K., Miller, C. C., Yantosca, R. M., Sulprizio, M. P., Thompson, A. M., Wennberg, P. O., Crounse, J. D., St. Clair, J. M., Cohen, R. C., Laughner, J. L., Dibb, J. E., Hall, S. R., Ullmann, K., Wolfe, G. M., Pollack, I. B., Peischl, J., Neuman, J. A., and Zhou, X.: Why do models overestimate surface ozone in the Southeast United States?, Atmos. Chem. Phys., 16, 13 561–13 577, doi:10.5194/acp-16-13561-2016, http://www.atmos-chem-phys.net/16/13561/2016/, 2016.

---

## Author Comment (AC2) · 14 Nov 2017

We thank the referee for their thorough review and thoughtful suggestions for improving the manuscript. Below are our responses to the referee's comments (*italics*).

*1 General comments*
*This manuscript presents a global uncertainty analysis of the concentration of oxidants*
*($O_3$, OH, and $HO_2$) at various altitudes through the troposphere, and in four geograph-*
*ical regions (central and northeastern U.S. and Canada, Gulf of Mexico, Pacific ocean*
*near Honolulu, Hawaii, and Pacific ocean near the southern coast of Alaska). The*

[Figure]

*authors use an ensemble of 512 GEOS-Chem simulations in which various inputs have been pseudo-randomly perturbed within prescribed uncertainty ranges, along with the high dimensional model representation (HDMR) technique to apportion the uncertainty in modeled oxidant concentrations to each of the perturbed inputs. The geographical regions studied in the manuscript feature various chemical and meteorological regimes and different local and upwind emissions profiles. Comparison of the results for these regions brings valuable insight into the model inputs that influence oxidant concentrations in these various conditions. The study is well conducted and the results are clearly presented and explained. A concern for the publication of this manuscript is the similarity of this study with a previous study (Christian et al., 2017a), mostly by the same authors. I do think that the proposed manuscript brings significant new contributions that warrant publication, but the authors should discuss more explicitly the insights that are novel and significant in this manuscript compared to the authors' previous work. The sections below describe in more detail the suggestions and comments that I would ask the authors to address prior to publication.*
*2 Specific comments*
*As mentioned in the Overview section, the authors should discuss the novel insights that this study brings compared to the previous work of Christian et al. (2017a). One novel and insightful aspect of the proposed manuscript seems to be the comparison of the uncertainty apportionment between different regions, as well as the vertical resolution of the analysis.*

Response: Compared to our previous study, there aren't too many differences in the methodology to highlight. As noted, we have presented the results in a slightly different format compared to the ARCTAS study with the sensitivities split vertically but this is more to highlight the vertically variable nature of these values. Perhaps the biggest difference between this study and the last is the inclusion of box model profiles for further comparison to the global model and measurements. Clearly the domains in this study are very different than the remote Arctic. These North American

and remote maritime domains are affected by a different set of local emissions and different chemical regimes.

Changes: Changed sentence to make note of difference between Christian et al., 2017a and this study (P6 L25).

*A similarly worded description of HDMR is already present in the previous work. Although it is useful for the proposed manuscript to summarize the principal concepts of this method, I suggest this description be re-worded further. The same comment applies to other parts of the "Methods" section.*

Response: We have expanded a bit on our methods section and the description of the HDMR method. Much of this is covered in our response to your later suggestions.

*Page 4, Equation (1): shouldn't $f_i(x_j)$ be $f_i(x_i)$ instead?*

Response: Yes, Changed as suggested. P4 Equation 1.

*Page 2, Lines 16-22: "Instead, the sensitivity analyses of GEOS-Chem modeled results has either used local methods in which the factor of interest is perturbed individually and compared to the model state without this perturbation, or the GEOS-Chem adjoint (Henze et al., 2007). [...] While useful in determining some individual sensitivities, these methods neither can nor were intentioned to provide a complete picture of model sensitivities in which many inputs have uncertainties." The adjoint sensitivity technique can be used to efficiently calculate first-order sensitivities of a model metric or cost function to many model inputs (sensitivities of a given model*

[Figure]

*metric or cost function to all model inputs can be efficiently calculated with a single "adjoint simulation"). Although the results from such an analysis provide "only" first-order local sensitivities, one can argue that they do provide a fairly comprehensive picture of model sensitivities for a given metric and a large number of model inputs that have uncertainties. A strength of the HDMR method used in the proposed manuscript resides in the fact that model non-linearities are accounted for in the propagation of uncertainties, while other sensitivity approaches are often limited to first-order sensitivities. However, the HDMR approach does require a large number of model simulations (512 here). Additionally, the apportionment of the overall uncertainty with the HDRM method relies on a priori estimates of the uncertainties on relevant inputs. Sensitivity or uncertainty apportionments based on other sensitivity methods often do not depend on such a priori estimates. Could the authors discuss these considerations in greater detail in the manuscript prior to publication?*

Response: It is a good idea to expand a bit on the strengths and weaknesses of this method in the context of the other sensitivity/uncertainty analysis methods currently used in the community. There are strengths and weaknesses in both the HDMR method and adjoint methods and there is quite a bit of overlap between the applications of adjoint and HDMR sensitivity tests, but there are some differences worth highlighting. For one, while adjoint models can be used to calculate model sensitivities, they are not necessarily used to determine the model uncertainty. Where the HDMR calculates the portion of the total model uncertainty attributable to the uncertainties in different factors, adjoint sensitivity tests do not put their sensitivities into the context of the total model uncertainty and do not fully sample the input space beyond some small perturbations. Secondly, the adjoint sensitivity tests can only be used for one output or cost function per test. In our cases we were looking at model-measurement agreement for multiple time periods during the field campaigns and for multiple outputs along these flight tracks. With our ensemble of model runs completed, we can easily compute sensitivities for any of a variety of different model outputs for any subset

of the field campaigns with negligible additional computational cost. Considering the differences in calculating adjoint sensitivities and the sensitivities calculated by the HDMR method, we hope that these results will be complementary to the work being done in the inverse modeling and adjoint community. We have reworded the introduction to make note of the work being done in the adjoint community and note the strengths and weaknesses of this method in comparison to the adjoint method.

Changes: Reworded third paragraph to mention and compare these methods to the adjoint sensitivity work. P2 L13-28

*Page 3, Lines 18–20: "As uncertainties are not published for the meteorological models, we define our meteorological uncertainties as the average of the monthly standard deviations of the difference between GEOS-4 and GEOS-5 meteorological fields for 2005, a year of overlap between the models." How different are these two models? If they are fairly similar, the uncertainties on meteorological inputs may be significantly underestimated. Can the authors discuss the fairness of this assumption?*

Response: The two models are more similar than dissimilar, but there are many differences between the two:
The native resolutions are different between the models (1x1.25 for GEOS-4, 0.5x0.666 for GEOS-5)
The data assimilation techniques used are different
The convective parameterizations are different
Cloud fraction and optical depth can be very different
http://wiki.seas.harvard.edu/geos-chem/index.php/Overview_of_GMAO_met_data_products#GEOS-4

With some back of the envelope calculations, we find similar uncertainties when
comparing the flight track temperatures between the model and measurements (∼2K vs 1.8K). Winds and some of the other meteorological factors perturbed in this study weren't measured on the aircraft. While our defining the uncertainties by the difference between models may result in some underpredictions, we are likely fairly close.

*Page 3, Lines 12-13: "In general, there were typically small differences between modeled results using either 4 x 5 or 2 x 2.5 resolutions but we illustrate in our results where this is not the case." The authors do discuss some of these differences in the Results section (for example: section 3.1.2), but it would be insightful to see more quantitative information describing the model-versus-observationss agreement (for example: mean bias, standard deviation) with the lower resolution simulations on the one-hand, compared to the higher resolution simulations on the other hand.*

Response: We did not discuss the differences between the coarse and fine results outside of the Houston flights because the differences were quite small compared to the differences seen among the different perturbed model runs. For those interested in the differences between these resolutions we have added a comparison of the two resolutions for each domain and added the fine resolution profiles to the supplement.

Changes: Added/expanded discussion of fine vs. coarse model resolution for each domain. P8 L5-7; P8 L19-23; P9 L6-8; P9 L27-20 and fine vs. coarse profiles to the supplement.

*Page 5, Equations (3) and (4): I am unsure as to whether $\varphi^i_r$ in equation (3) is the same as $\varphi^i_p$ and $\varphi^j_q$ in equation (4) if r = p = q and i = j. Additionally, the use of superscripts as indices can also introduce confusion. Can the authors add text to clarify these concepts?*

[Figure]

Response: We have added more descriptions of the what all the indices mean in the Methods section and added citations to direct readers to the papers that cover how some of these functions are created. The superscripts as indices follows convention established in previous HDMR papers. To lessen this confusion we have added some additional text to describe what these constants mean and removed the equation describing the calculation of the second order polynomials since we don't discuss these indices specifically anyways.

Changes: Various changes to the equations and their descriptions (P5).

*Can the authors discuss the contributions of uncertainties associated with inputs interacting with one another (i.e. "missing" slices in Figures 6–9), and the significance of these missing slices for the interpretation of the results presented in the manuscript?*

Response: The "missing" portion of the pie charts represent the portion of the total variance not accounted for by the variances of the first order sensitivity indices. One can think of this as some of the "non-linear" interactions between factors. While the software calculates these polynomials representing the co-varying of two factors at a time, we are not as confident in these values. Also these sensitivity indices are individually smaller than the first order indices. We are confident in our first order indices as we have tested the sensitivity indices calculated with varying numbers of model runs and find the sensitivity indices to converge upon a consistent value after 256 runs or so giving us confidence in the first order sensitivity indices presented here. (See Reviewer 1 Comment 1).

Changes: Made note of the missing portion of the pies on P10 L11-13

*3 Technical corrections: The authors repeatedly use the word "standard" to refer to notions such as "common practice" or "default value" or "default configuration". I suggest that the word "standard" be reserved for a more restrictive meaning of the word (i.e. a formalized norm or convention). Examples:*
*Page 3, Line 8: "We use in this study the standard GEOS-Chem model"*
*Page 3, Line 21: "the model ensemble made use of the standard emissions inventories"*
*Page 4, Line 7: "the standard model treatment"*
*Page 14, Line 21: "as opposed to the standard 0.20"*
*In what follows, text that I suggest be removed is written inside curly braces and in red, and suggested replacement text is in blue. Page 2, Line 15: "save for {a} some recent work"*

Changed as suggested

*Page 3, Line 8: "We use in this study the standard GEOS-Chem model (v9-02), a {popular} widely-used global chemical transport model"*
*Page 3, Line 21: "{For much of the developed world} For many industrialized regions"*
*Page 6, line 29: "During INTEX-A, the NASA DC-8 primarily sampled the eastern half of the United States and Canada {INTEX-A} during the summer of 2004"*
*Page 9, Line 5-6: "aerosol uptake {to} of $HO_2$"*
*Page 9, Lines 8-9: "In contrast, {uncertainty} uncertainties in both OH and $HO_2$ mixing ratios were considerable"*
*Figures 6-9: the different colors used in these Figures translate to very similar shades of gray when converted to gray-scale. I suggest changing some of these colors so that the different categories of inputs (Emissions, Kinetics, Photolysis, Meteorology, Heterogeneous) can be more easily distinguished when these Figures are converted*

*to gray-scale. I also suggest showing on these Figures the numerical values corresponding to the slices (i.e. contribution of each input to total uncertainty, in %), at least for the largest slices.*

Response and Changes: The colors were chosen using the ColorBrewer tool to optimize color viewing. To add some contrast, we have changed the colors of a couple of the categories. We have also taken the suggestion to add the sensitivity indices to the larger portions of the pie (any slice > 0.10).

*Acknowledgments: "University Maryland". Missing word "of"?*

Response: Correct. We thank the reviewer for finding this typo.

Changes: Changed as suggested (P15 L31)

———————————————————————